# The Usefulness of a Massive Open Online Course about Postural and Technological Adaptations to Enhance Academic Performance and Empathy in Health Sciences Undergraduates

**DOI:** 10.3390/ijerph182010672

**Published:** 2021-10-12

**Authors:** Rocío Martín-Valero, José-Manuel Pastora-Bernal, Lucía Ortiz-Ortigosa, María Jesús Casuso-Holgado, Veronica Pérez-Cabezas, Gema Teresa Ruiz-Párraga

**Affiliations:** 1Department of Physiotherapy, Faculty of Health Sciences, University of Malaga, C/Arquitecto Francisco Peñalosa 3, Ampliacion de Campus de Teatinos, 29071 Malaga, Spain; ortizolucia@gmail.com or; 2Department of Physiotherapy, Faculty of Health Science, University of Granada, 18071 Granada, Spain; jmpastora@ugr.es; 3Department of Physiotherapy, Faculty of Nursing, Physiotherapy and Podiatry, University of Seville, 41009 Seville, Spain; mcasuso@us.es; 4Department of Nursery and Physiotherapy, Faculty of Nursing and Physiotherapy, University of Cadiz, 11009 Cadiz, Spain; veronica.perezcabezas@uca.es; 5Department of Personality, Evaluation and Psychological Treatment, Faculty of Psychology, Institute of Biomedical Research of Malaga, University of Malaga, 29071 Malaga, Spain; gtruizparraga@uma.es

**Keywords:** empathy, MOOC, hypothetical model, learning, health sciences

## Abstract

Massive open online courses (MOOCs) provide accessible and engaging information for Physical Therapy and Occupational Therapy students. The objective of this research was to determine the usefulness in improving academic performance and empathy in health sciences undergraduates, and to test a hypothetical model through structural equation analysis. This research was carried out using a descriptive and quasi-experimental design. It was conducted in a sample of 381 participants: 176 used a MOOC and 205 did not. The results of the Student’s t-test showed statistically significant differences in academic performance between the groups in favor of those students who had realized the MOOC. Participants carried out an evaluation rubric after taking MOOC. Statistically significant differences in empathy were also obtained between the pre (X = 62.06; SD = 4.41) and post (X = 73.77; SD = 9.93) tests. The hypothetical model tested via structural equation modeling was supported by the results. Motivation for the MOOC explained 50% of the variance. The MOOC (participation and realization) explained 58% of academic performance, 35% of cognitive empathy and 48% of affective empathy. The results suggest an association between higher realization and participation in a MOOC and higher levels of academic performance, and cognitive and affective empathy.

## 1. Introduction

Massive open online courses (MOOCs) are intended to be (1) “massive” because thousands of students can access them; (2) “open” because participants do not pay any fees; (3) “online” because they are offered through the web and (4) “courses” because they are shaped around specific learning objectives by offering structured contents [1]. As McAuley put it, “a MOOC integrates the connectivity of social networking, the facilitation of an acknowledged expert in a field of study, and a collection of freely accessible online resources” [2]. MOOCs are considered very useful in increasing the educational levels of people who make use of them [3,4,5]. They use distance training that can include videos, exercises, presentations and assessments [3,4,5]. 

The use of MOOCs has increased due to the great opportunities they offer in the educational field [3]. MOOCs are increasingly used by health sciences and medical students [6]. This way of learning makes teaching easier, especially in the current COVID-19 crisis. Universities and colleges have been forced to use online formats, such as MOOCs [6]. Teaching through MOOCs offers benefits, such as great availability because they are online, free and accessible remotely. There is not much evidence of what students achieve through MOOCs [4]. Another problem regarding MOOCs is that there is no certainty as to the best type of tools to evaluate them [3].

Numerous scientific publications based on MOOCs have been published during the last years. Many of them have focused on health science students [7,8,9]. A review from 2014 suggests that MOOCs can be used as a way to provide continuous medical education [10]. It also shows the potential of MOOCs as a means to increase health literacy among the public. A recent study confirmed that students’ academic performance can be influenced by MOOCs which have the benefit of facilitating the learning process by offering materials and enabling information sharing [11]. Motivation is another aspect that has been briefly studied regarding the use of MOOCs, particularly in health science students [12]. Finally, the concept of empathy has been addressed in some studies with promising preliminary results [13]. 

Emotional intelligence (EI) is a relevant trait for students, particularly in health sciences and medicine [14]. EI is defined as the ability to understand and direct both one’s own emotions and those of other people. This is linked to an increase in empathy, which is why empathy is a key variable within EI [14,15]. Empathy is considered a very important quality both for students and health workers [14,16]. There are two types of empathy. Both involve the ability to understand other people in relation to the context in which they find themselves (cognitive empathy) or their emotions (affective empathy) [14]. Greater amounts of empathy in a person have been observed to lead to greater peace of mind and less discomfort. Improvements have also been found in the level of treatment adherence and results. On the other hand, it has been perceived that low levels of empathy can lead to a greater number of errors [14]. Empathy is, therefore, considered an important variable in health workers and students, which is why their instruction is critical to their education [16]. 

In recent times, technologies have become mainstream, both in people’s daily lives and in teaching. An example of this is the use of virtual reality (VR) to teach empathy to health professionals (doctors, physiotherapists and nurses), and the results of the use of VR show that it increased the empathy of students towards their patients [17]. Another example may be the use of artificial intelligence (AI) in healthcare. Studies show that AI will make the work of health workers easier so that they will have more time to devote to each patient, which will increase empathy [18]. 

To our knowledge, no other studies have investigated the relationship between MOOC use and academic performance, cognitive and affective empathy and motivation. Therefore, we think that the research on the subject is very important. We use MOOCs in order to seek an effective innovation strategy for the training of Physical Therapy and Occupational Therapy students. The aims of this study were: (a) to assess the usefulness of a MOOC to improve the academic performance of students enrolled in a Degree in Occupational Therapy and a Degree in Physical Therapy; (b) to test whether taking the MOOC increases empathy scores, through a pre-post analysis and (c) to test a hypothetical model through structural equation analysis, on the participants who realized a MOOC. In addition to this, we wanted to check how motivation contributes to the engagement in MOOCs, MOOC completion, academic performance (total score attained by the subject) and to cognitive and emotional empathy. The hypothesis was that motivation to undertake a MOOC should have a direct effect on participation in the MOOC, and MOOC completion. We predicted a positive association between MOOC and academic performance, as well as cognitive and emotional empathy.

## 2. Materials and Methods

### 2.1. Study Design and Participants

This research was carried out using a descriptive and quasi-experimental design. The sample consisted of students from the second and third year of the Degree in Occupational Therapy and the fourth year of the Degree in Physical Therapy at the University of Malaga. Inclusion criteria were students of the aforementioned subjects and good mastery of the Spanish language. Measurements were taken during the Academic Year 2019/2020. 

### 2.2. Ethical Considerations

Ethical approval for the educational research in this study was obtained from the Andalucía Ethics Committee on Human Research (PEIBA nº 7/2020_PI1). Informed consent was signed by all participants prior to educational intervention, following the recommendations of the Declaration of Helsinki. This study was conducted in accordance with the Research Ethics for Future Learn guidelines [19].

### 2.3. Study Procedures

A teaching activity was carried out in the following five subjects: “Support products, ergonomics and autonomy”, “Occupational therapy for personal autonomy in mental health III”, “Psychopathology in mental health”, “Analysis of activity in occupational therapy”, and “Psychological intervention in pain and hospitalization”. Each teacher explained the study to their group of students and offered them the opportunity to carry out an extension activity on the acquisition of professional and personal skills. They were provided with a web link to a course entitled “Postural and Technological Adaptations in Pediatrics" hosted on the MiriadaX platform [5,20]. The following six modules were presented in the MOOC theme: the concepts of disability and their evaluation, and possible postural adaptations that can be designed to address therapeutic intervention in pediatrics and show technological tools available as aid. Students also learned to design and build an assistive product with the help of the low-cost philosophy. The last module facilitates learning based on "adapted play" where individual adaptations are created for people with functional diversity [5]. 

Videos of the MOOC were selected, where students were made aware of the possibilities of postural adaptations that can be designed to address therapeutic intervention in pediatrics. In each module of the MOOC, there was a brief theoretical explanation with a discussion of clinical cases, analysis and clinical reasoning in response to a problem raised. Videos of short duration, between two and three minutes, were proposed to build each unit, in which the content is developed. Students could view various clinical cases throughout the development of the course and had the opportunity to participate in discussion forums to increase engagement and interaction. 

Participants realized an online evaluation protocol before and after taking the MOOC. This protocol included sociodemographic and academic data and psychological variables, such as motivation and empathy. At the end of the MOOC, participants realized a MOOC evaluation rubric that is available online (Appendix A). The rubric consisted of 11 items describing the skills that students must achieve at the end of the course. These competencies are attitudinal, conceptual and procedural in character, based on the objective that students should achieve. Depending on the level of knowledge and learning skills, students obtained a score of 0 to 3 points for each competence. “0” represented the minimum acceptable knowledge and “3” the highest level of skills acquired. The final score was from 0 points to a maximum of 33 points at the end of the self-assessment [21]. 

### 2.4. Outcome Measures

#### 2.4.1. Demographic Variables

Participants were asked to provide information on age, gender, type of university degree, and current year of study.

#### 2.4.2. Empathy

The Spanish version of the Basic Empathy Scale was used to assess this variable [22]. The scale consists of 20 items (i.e., I quickly notice when a friend is angry) and uses a 5-point Likert scale answer format, ranging from 1 (strongly disagree) to 5 (strongly agree). Nine items are related to cognitive empathy (BES-C) and 11 items are related to affective empathy (BES-A). A higher score indicates greater empathy. The BES had a good discriminant and convergent validity with regard to measurements of Narcissism, Psychoticism and Agreeableness [22].The BES showed suitable reliability for the sample of the present study (Cronbach´s alpha = 0.83).

#### 2.4.3. x-MOOC (Completion and Participation)

Completion of the MOOC was assessed with a questionnaire at the end of each of the six modules (0–100 points). The maximum score that could be obtained was, therefore, 600 points. Participation in forums and activities was scored as a maximum of 150 points.

#### 2.4.4. Academic Performance

This variable was obtained using the total score in the subject, which included the exam results, resolution of practical cases and final project. The maximum possible score was 10 points.

#### 2.4.5. Motivation to Join a MOOC

This variable was assessed with four items: 1. Motivation is an important factor in achieving consistency in the teaching-learning process 2. You are motivated to complete this online training. 3. You think that having fun is important for learning. 4. You believe that consistency benefits academic performance. A 3-point Likert scale was developed ad-hoc using expert criteria. A high score indicates greater motivation to join a MOOC. The items showed suitable reliability for the sample of the present study (Cronbach´s alpha = 0.75). 

#### 2.4.6. Statistical Analysis

Statistical analyses were conducted using the SPSS (Windows version 26.0, SPSS Inc., Chicago, IL, USA) and AMOS Graphics (version 26.0; Small Waters Corp., Chicago, IL, USA) software packages.

Sample size was calculated using the G*Power software (version 3.1.2, Kiel University, Kiel, Germany). We assumed a two-tailed hypothesis, an equal distribution of participants in the study groups, a medium effect size (d = 0.47) [8], an alpha level of 0.05, and a 99% power. Considering a 10% dropout rate, a total of 368 participants were required for the study. Finally, 381 students were recruited.

The data were first examined for incomplete responses, within-groups Mahalanobis distance, and the assumptions of normality and homoscedasticity. To study the differences in the change of means in the academic performance obtained between the students who participated in the x-MOOC and the group who did not participate in the x-MOOC, the Levene test of equality of variances and the t-student test for the differences of means were used in two independent samples if the validity conditions were met. The power of the effect size was calculated with Cohen’s d. The numerical variables were described with mean and standard deviation and the qualitative variables by frequency and percentages. Pearson´s correlations were then calculated for each continuous variable measured in the study. The t-test for related samples was applied to see if there were differences between the means in pre-post test empathy. In all tests, results were considered statistically significant when alpha values were <0.05, at a 95% confidence interval.

The hypothetical model was tested via SEM. All analyses used maximum likelihood estimation and robust estimation methods. In line with the recommendations of two books [23,24], model fit and convergence between findings were analyzed using several goodness-of-fit indexes: the Satorra–Bentler chi-square, the root mean square error approximation (RMSEA), the goodness-of-fit index (GFI), the adjusted goodness of fit index (AGFI), and the comparative fit index (CFI). The Satorra–Bentler chi-square is a fit index that corrects the statistic under distributional violations [25]. RMSEA values less than 0.08 indicate an adequate fit. Regarding the GFI and AGFI, the closer the values are to 1 the better the fit; higher values indicate well-fitting models. The CFI measures the proportional improvement in fit by comparing a hypothesized model with a more restricted baseline model. The CFI index ranges from 0 (absolute lack of fit) to 1 (perfect fit). It is generally accepted that values equal to or more than 0.95 in these goodness-of-fit indexes indicate well-fitting models.

Five observable variables or indicators of the latent variables were used. Four latent variables (motivation to join a MOOC, MOOC (completion and participation), academic performance, cognitive empathy, and affective empathy were associated in the hypothesized structural equation model. Motivation to join a MOOC was specified by four items, MOOC (completion and participation) was specified by the total score in the subject and total score for participation, cognitive empathy was specified by BES-C sub-scale and affective empathy by the BES-A sub-scale according to with previous confirmatory analyzes ([χ2 (df = 26) = 26.789], RMSEA = 0.015, TLI = 0.99, CFI = 0.99). One loading for each latent variable was fixed at 1.0 for setting the metric of the latent construct.

## 3. Results

### 3.1. Preliminary Analysis

The total sample consisted of a total of 381 participants, of whom 176 took the MOOC and 205 were part of the control group. The participants were mostly women (75%) and with a mean age of 21 (SD = 1.56) years.

Based on the results of the preliminary analysis, two participants who were multivariate outliers were excluded from data analyses (Mahalanobis distance *p* < 0.001; [26]. The assumptions of normality and homoscedasticity were confirmed. There were no statistically significant differences in terms of age and sex between the experimental group and the control group (with MOOC and without MOOC, respectively) in the five subjects (see Table 1). Furthermore, correlations between variables did not indicate any associations greater than 0.90, a Durbin Watson statistic greater than 4, or other problems associated with multicollinearity or homoscedasticity.

### 3.2. Usefulness of the MOOC to Improve the Academic Performance of Students Enrolled in a Degree in Occupational Therapy and the Degree in Physical Therapy

The corresponding comparisons of means were carried out in the scores obtained by the Student’s t-test for independent samples. The results (Table 2) showed statistically significant differences in academic performance obtained between both groups in the four subjects whose content was related to the MOOC, in favor of those students who had realized the MOOC. The students who took the subject whose content was not closely related to the MOOC also achieved higher academic performance, but this difference did not reach statistical significance. The results that showed statistically significant differences showed a medium or high effect size power.

The results of the MOOC evaluation rubric carried out by the participants of the experimental group were quite good (X = 21; SD = 6.88, out of a maximum of 36 points).

### 3.3. MOOC and Empathy: A Pre-Post Analysis

The analysis of the mean comparison t-test for related samples obtained statistically significant differences between the pre- (X = 62.06; SD = 4.41) and post- (X = 73.77; SD = 9.93) empathy scores (X = –11.71; SD = 11.36; t(173) = –13.68, *p* < 0.001) with a high effect size size (d = 0.86). 

### 3.4. Bivariate Analyses

We calculated bivariate correlations between motivation to join a MOOC, MOOC (completion and participation) academic performance, cognitive empathy, and emotional empathy. Table 3 shows the descriptive statistics (means and standard deviations) and correlations of the measures used in the structural equation analysis.

All the correlations between the variables (see Table 3) were significant (*p* < 0.001), and were in the expected direction. 

### 3.5. The MOOC’s Contribution to Academic Performance and Cognitive and Emotional Empathy: Structural Equation Model

The overall pattern of results broadly supported the hypothetical model. Moreover, the assessment of the model indicated a good fit for the data. The relative chi-square for the model was suitable [χ^2^ (df = 2, *n* = 173) = 5.291, *p* = 0.158], the RMSEA was 0.04, and the CFI, GFI, and AGFI values were all equal to 0.99. Figure 1 shows the final model with standardized coefficients and R2 values. 

According to the results, motivation to join a MOOC yielded a statistically significant path coefficient to MOOC (completion and participation), explaining 50% of the variance of this variable. The results suggest an association between higher levels of motivation to join a MOOC and higher levels of completion and participation in the MOOC. The x-MOOC yielded a statistically significant path coefficient to academic performance (explaining 58% of the variance of this variable), to cognitive empathy (explaining 35% of the variance of this variable) and to affective empathy (explaining 48% of the variance of this variable). The results suggest an association between higher completion and participation in the MOOC and higher levels of academic performance, and cognitive and affective empathy.

## 4. Discussion

This study shows the results obtained regarding academic performance and empathy in students enrolled on a Degree in Occupational Therapy and a Degree in Physical Therapy, after carrying out an educational intervention with a MOOC as an extension activity within their university training. A direct association was found with the participation in the MOOC, academic performance and empathy increase. Moreover, the motivation to join a MOOC appeared to be related to higher levels of academic performance and empathy. 

### 4.1. MOOCs Assessment Methods

Systematic reviews recommend that the quality assessment of the MOOC be performed through guided peer review with rubrics [27]. A recent systematic review recommends that there are assessment strategies and multi-learning methods adapted to the needs of the attendees [28]. Two of the articles mentioned in the review use the rubric as an assessment method in medical and nursing students [9,29]. The rubric has also been used in this research as an evaluation method, including the most important sections of the MOOC so that students can carry out a self-evaluation of organization, knowledge and encouragement to participate in the discussion forums during the MOOC [21]. The students responded to the questions posed in the online rubric [21]. However, there are many types of rubrics that may be used to evaluate MOOCs. A previous study used a rubric consisting of comprehension questions, and others focused on the reflection area [30]. The rubric score was divided into two levels, where a score from 4 to 6 points means success and from 1 to 3 means failure [30]. A more specific rubric was similarly used in the field of science and technology to perform a peer assessment of student learning and motivation in an astronomy MOOC [31]

### 4.2. Usefulness of the MOOC to Improve the Academic Performance of Students Enrolled in the Degree in Occupational Therapy and the Degree in Physical Therapy

Our research has shown an increase in the academic performance of participants who had realized the MOOC educational intervention with respect to the group who did not realize the MOOC. Previous studies have used the Kirkpatrick model to evaluate the qualitative synthesis of the effect of MOOCs students through deductive thematic analyses [4]. The Kirkpatrick model uses the following four levels: reaction, learning, behavior and results [4]. Gains in the students’ learning levels were observed through surveys developed to assess the effect of the MOOC on knowledge; however, they did not ascertain with certainty how the students benefited from the MOOC [4]. 

### 4.3. MOOC and Empathy: A Pre-Post Analysis

Previous studies showed that empathy is a skill that may be taught [13,15,32,33,34]. Our study has also suggested that emotional competencies can be learned while training for health professions with a MOOC. For empathy, we used the Spanish version of the BES scale in our study, according to the age of the study population. Previous research used the Italian version of the Balanced Emotional Empathy Scale (BEES) in student nurses during a three year degree [34]. Other research assessed emotional intelligence and empathy through two validated scales: the Schutte Self-Report Emotional Intelligence Test (SSEIT) and the Jefferson Scale of Empathy-Health Professions Student Version (JSE-HPS) [15]. The first one (SSEIT) contains 33 items that measure the expression and evaluation, regulation and manipulation of emotions (Cronbach’s alpha 0.73–0.92). The second scale (JSE-HPS) checks for three factors: “perspective taking”, “compassion care” and “standing in the patient’s shoes” (Cronbach’s alpha 0.81) [15]. Unlike previous studies that evaluated empathy using the interpersonal reactivity index (IRI), it is made up of four subscales, "perspective taking, empathic concern, personal anguish and fantasy" of seven items, which measure both cognitive and affective empathy [14]. In this study, Cronbach’s alpha was 0.716 for perspective taking, 0.725 for empathic concern, and 0.659 for personal distress [14]. In contrast, the Cronbach’s alpha of the BES scale was 0.83, which is higher. Therefore, our research also confirms the literature data on the reliability of the empathy scales that obtained a Cronbach´s alpha coefficient above 0.8.

### 4.4. The MOOC’s Contribution to Academic Performance and Cognitive and Emotional Empathy: Structural Equation Model

This is the first study in which a hypothetical model relates the performance of, and engagement in a MOOC to motivation, academic performance, and cognitive and affective empathy. To our knowledge, no previous studies involve a hypothetical model to connect these dimensions in health sciences students. 

Motivation was measured through questions on an online self-assessment questionnaire with a Cronbach’s alpha coefficient = 0.75. A previous study identified the learners’ motivation and engagement as affecting the completion of a MOOC [12]. To measure motivation, another study created a motivation scale based on the idea of a type of motivation [35]. Another study measured the level of motivation by asking the study subjects to complete the Self-Regulation of Learning Questionnaire [33]. The questionnaire contained statements related to autonomous and controlled motivation. Cronbach’s alpha was 0.80 for autonomous motivation, and 0.75 for controlled motivation [36]. In another investigation, motivation was measured through two questionnaires that were administered online (one was aimed at teachers and the other at students) [37]. The questionnaires assessed the way in which online education was delivered for both teachers and students, whether teachers and students found this type of teaching for dental learning useful, and finally motivation for learning online. This was measured using a 5-point Likert scale The Cronbach’s alpha for educational benefit was 0.659, and that the value for the management of pooled data was 0.729 [37]. 

The results of our study suggest a relationship between higher rates of engagement in and completion MOOCs and higher levels of academic performance. Another study made a hypothetical model where one of the hypotheses was that the satisfaction shown by a student would result in more efficient use of the MOOC. The primary tool used by the study to carry out the analysis was a structural equation model. The hypothesis was accepted since the relationship between student satisfaction and effectiveness was significant β = 0.809, *p* < 0.001. Therefore, the study finally confirms that the use of the MOOC can affect the effectiveness of academic performance because it favors learning thanks to the exchange of information and the offer of resources [38]. 

### 4.5. Implications and Limitations 

Our results have some implications for teaching and learning. First, MOOCs are free tools that may help university students to acquire some additional skills. Second, this online training seems to have a positive correlation with academic performance. Furthermore, this tool promotes both cognitive and emotional empathy, which are important skills in health sciences degrees. Finally, it is also worth noting that the use of the rubric that we suggested is a useful tool to assess the MOOC. Our study has, however, some limitations. First, it is a quasi-experimental study with small sample size. Second, the participants were not randomly assigned s to groups. Third, we used a non-validated questionnaire to measure the motivation. It is recommended to increase the sample size and include a control group and the randomization of participants. 

## 5. Conclusions

The results suggest an association between academic performance and participation in MOOCs. Cognitive and emotional empathy also improved after MOOC training. A structural equation model reported that higher realization and participation in a MOOC were related to higher levels of academic performance, and cognitive and affective empathy. Thus, this study highlights the importance of using MOOCs as an adjuvant tool to teach academic and professional skills in occupational therapy and physical therapy. MOOCs can be a promising way to develop empathy in health education.

## Figures and Tables

**Figure 1 ijerph-18-10672-f001:**
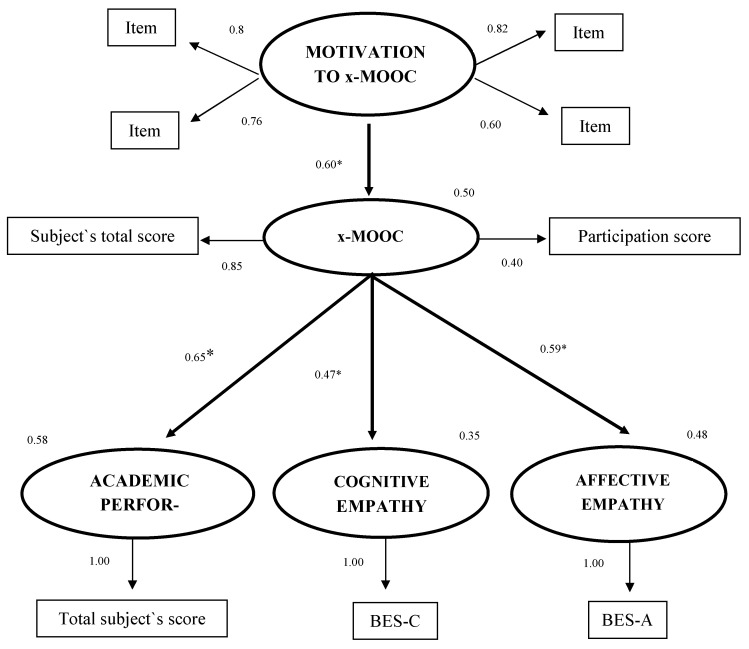
Empirical model. Observed variables (Factors/Scales) are represented by squares and latent variables by circles. Abbreviations: BES-C = Cognitive sub-scale of the Basic Empathy Scale; BES-A = Affective sub-scale of the Basic Empathy Scale * *p* ≤ 0.01.

**Table 1 ijerph-18-10672-t001:** Preliminary analysis: analysis of differences in age and sex between the groups without and with MOOC in 5 subjects of the Degree in Occupational Therapy and Physical Therapy.

Subjects	Age		Sex
	*n*	*t*-Student	Chi-Square Pearson Tests
Activity analysis	99	*t*(99) = 0.49, *p* = 0.485	*χ^2^* (1, *n* = 99) = 0.171, *p* = 0.445
Personal autonomy in mental health	108	*t*(108) = –0.808, *p* = 0.212	*χ^2^* (1, *n* = 108) = 1.03, *p* = 0.500
Psychopathology in mental health	36	*t*(36) = 0.187, *p* = 0.421	*χ^2^* (1, *n* = 36) = 0.778, *p* = 0.486
Support products, ergonomics, and personal autonomy	107	*t*(107) = 0.080, *p* = 0.936	*χ^2^* (1, *n* = 107) = 0.271, *p* = 0.400
Pain and hospitalization	29	*t*(29) = –0.247, *p =* 0.807	*χ^2^* (1, *n* = 29) = 0.343, *p* = 0.453

**Table 2 ijerph-18-10672-t002:** Descriptive statistics and comparison of means in grades between the groups without and with MOOC in five subjects of the Degree in Occupational Therapy and the Degree in Physical Therapy.

Subjects	Without x-MOOC (Control Group)			With x-MOOC (Experimental Group)			*t*-Student	Cohen’s *d*
	*n*	X	SD	*n*	X	SD		
Activity analysis	48	6.93	1.34	51	7.45	0.68	*t*(99) = –2.38, *p* < 0.001	0.51
Personal autonomy in mental health	60	6.85	1.66	48	7.38	1.56	*t*(108) = –1.68, *p* < 0.001	0.53
Psychopathology in mental health	18	7.06	1.41	18	7.26	0.95	*t*(36) = 0.483, *p* = 0.310	0.20
Support products, ergonomics, and personal autonomy	61	7.21	1.26	48	7.75	1.00	*t*(107) = –2.40, *p* = 0.015	0.50
Pain and hospitalization	18	8.71	1.92	11	9.39	1.76	*t*(29) = -2.01, *p* = 0.040	0.66

**Table 3 ijerph-18-10672-t003:** Descriptive statistics (means and standard deviation) and bivariate correlations of the variables in the experimental group (*n* = 173).

Variables	Mean (SD)	1	2	3	4	5
1. Motivation to MOOC	4.35 (0.57)	1				
2. MOOC	421.65 (95.3)	0.68 *	1			
3. Academic performance	7.59 (1.20)	0.38 *	0.77 *	1		
4. Cognitive empathy	33.19 (4.46)	0.51 *	0.74 *	0.51 *	1	
5. Emotional empathy	40.57 (5.46)	0.59 *	0.76 *	0.55 *	0.88 *	1

* significance *p* < 0.001.

## Data Availability

The data that support the findings of this study are available from the corresponding author upon reasonable request.

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
