# Peer review of "The Usefulness of a Massive Open Online Course about Postural and Technological Adaptations to Enhance Academic Performance and Empathy in Health Sciences Undergraduates"

_ijerph, 2021, doi:10.3390/ijerph182010672_

Round 1
Reviewer 1 Report
In “The usefulness of the massive open online courses postural and technological adaptations to enhance academic performance and empathy in health sciences undergraduates,“ the authors study the impact on students who use MOOCs to supplement their learning versus those that do not. They found a positive impact of MOOC participation on academic performance and empathy. Those students motivated to take the MOOC were more likely to complete the MOOC option. This work contributes a valuable case to demonstrate positive outcomes to MOOC participation in a quantitative manner, which is limited in the field. There is also value in showing that empathy improves through taking a MOOC, a skill that some may be skeptical of being achieved through an online course.
The revisions helped address the gaps in the introduction in the original submission and highlighted the limitation of the experimental design.
The document still requires more proofreading for English grammar (entire document). Some examples include subject-verb agreement (line 72 of the track change document: suggest to suggests and line 525 MOOC to MOOCs) or missing words like in the first line of the abstract (MOOCs based pediatric has to MOOCs based on pediatrics have.) Thorough editing should catch all of the cases.
The first and last lines of the current introduction repeat some information.
Overall, I think this paper makes a positive contribution to the field and should be published.
Author Response
ITEMIZED LIST OF REVIEWER´S COMMENTS
Reference: [ijerph-1387545]
Title: "The usefulness of a massive open online course about postural and technological adaptations to enhance academic performance and empathy in health sciences undergraduates".
Dear Reviewer,
We greatly appreciate the editor´s and reviewers’ kind and encouraging comments about our study. We have followed their suggestions, trying to incorporate them into the revised version of our manuscript. We uploaded the tracked changes manuscript, the clean version revised manuscript and itemized point-by-point response to the reviewer’s comments are presented below.
Editor´s and Reviewers´ comments:
*Reviewer 1
RV: Reviewer
AA: Authors
RV: GENERAL COMMENTS:
In “The usefulness of the massive open online courses postural and technological adaptations to enhance academic performance and empathy in health sciences undergraduates,“ the authors study the impact on students who use MOOCs to supplement their learning versus those that do not. They found a positive impact of MOOC participation on academic performance and empathy. Those students motivated to take the MOOC were more likely to complete the MOOC option. This work contributes a valuable case to demonstrate positive outcomes to MOOC participation in a quantitative manner, which is limited in the field. There is also value in showing that empathy improves through taking a MOOC, a skill that some may be skeptical of being achieved through an online course.
The revisions helped address the gaps in the introduction in the original submission and highlighted the limitation of the experimental design.
AA: First, authors want to thank the modifications suggested by the reviewer and his/her effort to improve our manuscript. Following the reviewer’s recommendation, we have improved our manuscript. Thank you for your comments.
RV: The document still requires more proofreading for English grammar (entire document). Some examples include subject-verb agreement (line 72 of the track change document: suggest to suggests and line 525 MOOC to MOOCs) or missing words like in the first line of the abstract (MOOCs based pediatric has to MOOCs based on pediatrics have.) Thorough editing should catch all of the cases.
AA: Following the reviewer’s recommendation, a spelling and grammatical revision of the text was performed. Please find attached the certificate of the translation service (at the end of this letter).
First, we have corrected mistakes in line 56 on page 2 in the revised manuscript. Secondly, we have also corrected in line 430 on page 10 in the revised manuscript. Finally, we have improved in the first sentence of the abstract.
It is explained in the revised manuscript as follows below:
“Massive open online courses (MOOCs) provide accessible and engaging information for Physical Therapy and Occupational Therapy students”.
RV: The first and last lines of the current introduction repeat some information.
AA: We agree with you. Thank you for your comment. We have improved in lines 43 to 45 on page 1 in the revised manuscript. It is explained in the revised manuscript as follows below:
“They use distance training that can include videos, exercises, presentations and assessments”.
I would like to take this opportunity to inform you that all references have been reviewed and updated throughout the manuscript.
RV: Overall, I think this paper makes a positive contribution to the field and should be published. AA: Without a doubt, their recommendations help us to give more clarity and structure to our manuscript. Thank you very much for your suggestions.
All the best

Reviewer 2 Report
The present study was to determine the utility of improving academic performance and empathy in health sciences undergraduates, and to test a hypothetical model through structural equation analysis. Although the manuscript has been revised, in my opinion, the manuscript still needs a lot of attention. Thus, I just have some suggestions.
- The introduction needs a stronger justification for the purpose of this study. It is a lengthy introduction, but the justification for the study needs to be stronger.
- Please add a conceptual framework.
- Please include the sample size calculation.
- Table 1 is not appropriate to be put under Methods, it should be included as an appendix or supplemental file. Authors can list out the items’ names in text rather than provide the questionnaire in a Table. The questionnaire in table format should be in the Appendix.
- The CFI value should be more than 0.95 to indicate the model fits the data well. Please provide the latest citation for your cut off point 0.90.
- In Table 4, the correlation should include cognitive empathy and affective empathy.
- Please provide the data collection date.
- The discussion should be based on comparisons with previous studies, rather than solely on the results.
- Overall, I felt the study need stronger justification, conceptual framework etc. Thank you. All the best.
Author Response
ITEMIZED LIST OF REVIEWER´S COMMENTS
Reference: [ijerph-1387545]
Title: "The usefulness of a massive open online course about postural and technological adaptations to enhance academic performance and empathy in health sciences undergraduates".
Dear Reviewer,
We greatly appreciate the editor´s and reviewers’ kind and encouraging comments about our study. We have followed their suggestions, trying to incorporate them into the revised version of our manuscript. We uploaded the tracked changes manuscript, the clean version revised manuscript and itemized point-by-point response to the reviewer’s comments are presented below.
Editor´s and Reviewers´ comments:
*Reviewer 2
RV: Reviewer
AA: Authors
RV: GENERAL COMMENTS:
The present study was to determine the utility of improving academic performance and empathy in health sciences undergraduates, and to test a hypothetical model through structural equation analysis. Although the manuscript has been revised, in my opinion, the manuscript still needs a lot of attention. Thus, I just have some suggestions.
RV: 1. The introduction needs a stronger justification for the purpose of this study. It is a lengthy introduction, but the justification for the study needs to be stronger. 2.Please add a conceptual framework
AA: Following the reviewer´s recommendation, we have added new sentences to justify our publications.
A conceptual framework was added in lines 87 to 89 on page 02 in the revised version in the introduction section: “We use MOOCs in order to seek an effective innovation strategy for the training of Physical Therapy and Occupational Therapy students”.
Thank you for your suggestion.
- 3: Please include the sample size calculation
AA: Indeed, we had forgotten to include this information. The sample size calculation has been included in lines 185 to 289 on page 4 in the revised version.
- 4. Table 1 is not appropriate to be put under Methods,it should be included as an appendix or supplemental file. Authors can list out the items’ names in text rather than provide the questionnaire in a Table. The questionnaire in table format should be in the Appendix.
AA: We agree with you. Following the reviewer´s recommendation, the questionnaire has been included such as a supplementary Table: “Table S1: Description of Assessing a Massive Online Course Rubric”. Therefore, we have updated the numbering of the tables in the manuscript.
- 5. The CFI value should be more than 0.95 to indicate the model fits the data well. Please provide the latest citation for your cut off point 0.90.
AA: We thank you for your observation. This data has been modified on page 5.
- 6. In Table 3, the correlation should include cognitive empathy and affective empathy.
AA: We agree with your proposal. We have included in Table 3 the data of both types of empathy.
- 7. Please provide the data collection date.
AA: Indeed, we had forgotten to include this information in the method section. It is explained in lines 106 to 107 on page 3 in the revised manuscript as follows below:
“Measurements were taken during the Academic Year 2019/2020”.
RV: 8. The discussion should be based on comparisons with previous studies, rather than solely on the results.
AA: We agree with you. Following the reviewer´s recommendation, we have remade new sentences in lines 382 to 400 on page 10 in the revised version:
“Previous studies showed that empathy is a skill that may be taught [13,15,32–34]. Our study has also suggested that emotional competences can be learned while training for health professions with a MOOC. For empathy, we used the Spanish version of the BES scale in our study, according to the age of the study population. Previous research used the Italian version of the Balanced Emotional Empathy Scale (BEES) in student nurses during a three years degree [34]. Other research assessed emotional intelligence and empathy through two validated scales: the Schutte Self-Report Emotional Intelligence Test (SSEIT) and the Jefferson Scale of Empathy-Health Professions Student Version (JSE-HPS) [15]. The first one (SSEIT) contains 33 items that measure the expression and evaluation, regulation and manipulation of emotions (Cronbach's alpha 0.73-0.92). The second scale (JSE-HPS) checks for three factors: “perspective taking”, “compassion care” and “standing in the patient's shoes” (Cronbach's alpha 0.81) [15]. Unlike previous studies that evaluated empathy using the interpersonal reactivity index (IRI), it is made up of four subscales, "perspective taking, empathic concern, personal anguish and fantasy" of seven items, which measure both cognitive and affective empathy [14]. In this study, Cronbach's alpha was 0.716 for perspective taking, 0.725 for empathic concern, and 0.659 for personal distress [14]. In contrast, the Cronbach’s alpha of the BES scale was .83, which was higher. Therefore, our research also confirms the literature data on the reliability of the empathy scales that obtained a Cronbach´s alpha coefficient above 0.8”.
I would like to take this opportunity to inform you that all references have been reviewed and updated throughout the manuscript.
RV.9. Overall, I felt the study need stronger justification, conceptual framework etc. Thank you.
AA: Without a doubt, their recommendations help us to give more clarity and structure to our manuscript.
Please, do not hesitate to contact me, if you require further corrections. Thank you in advance
All the best

This manuscript is a resubmission of an earlier submission. The following is a list of the peer review reports and author responses from that submission.
Round 1
Reviewer 1 Report
In “The usefulness of the massive open online courses postural and technological adaptations to enhance academic performance and empathy in health sciences undergraduates,“ the authors study the impact on students who use MOOC to supplement their learning versus those that do not. They found a positive impact of MOOC participation on academic performance and empathy. Those students motivated to take the MOOC were more likely to complete the MOOC option. This work contributes a valuable case to demonstrate positive outcomes to MOOC participation in a quantitative manner, which is limited in the field. There is also value in showing that empathy improves through taking a MOOC, a skill that some may be skeptical of being achieved through an online course.
I do have some concerns and suggestions for revision. I am concerned about the quasi-experimental design of the study and conclusions as the authors list as a limitation of the study. If I understand it correctly, they gave students the option to participate in the MOOC to enhance their studies. So the students who did participate in the MOOC did better. The two populations (using MOOCs versus not using the MOOCs) are not random or equal presumably in terms of motivation and academic achievement before the intervention. To put it another way, if the authors just simply asked the students if they would want to participate in a MOOC with additional materials, would they see the same result for the academic performance of those who said yes versus no without the students even using the materials? They are self-selecting, which seems like a major bias to see the results as hypothesized for academic performance. Those who opt in to do more will do better.
The authors state, “Another problem that MOOCs present is that because they are very new it is not known with certainty the type of tools that are the best used to evaluate them [1].” I would not consider the xMOOCs very new. They have been around since at least 2011 and have a lot of research published about them. Beyond that, there are decades of research involving online courses or blended learning that pull in concepts similar to this paper.
The introduction should be re-written to be much more sophisticated in the area of MOOC research. They must add in the papers tying together the topics of MOOCs and use with health science students, academic performance, empathy, or motivation that the authors say do not exist, but many examples are tying MOOCs to any of these topics on their own but maybe not in that complete combination.
Overall, the writing should be thoroughly checked for grammar and clarity. I found many problems, but here are a few specific suggestions.
I find the term b-MOOC odd because it is rarely used. If the authors are going to use it, they should cite an original definition and not their work using the term.
Remove calling the MOOC x-MOOCs all the time. The x versus c is rarely used.
The abstract and first two paragraphs of the introduction are very choppy with short sentences. The sentences don’t flow well and sometimes repeat with information that contributes little to the point of the study. The authors mostly use passive voice instead of active voice in the abstract.
In the abstract, this line provides little value. “This research was based on x-MOOCs that have more classical form.” Replace this line with specifically what type of course the authors used and why.
Missing “that” in line 44.
Line 46, should be “MOOCs”
Line 86 does not make sense grammatically.
Table 3 is problematic in the formatting.
Author Response
ITEMIZED LIST OF THE REVIEWERS COMMENTS
Reference: [ijerph-1281527]
Title: Postural and technological adaptations of massive open online courses in pediatrics: Utility in improving academic performance and empathy
Dear Reviewer,
The authors want to thank your thoughtful and constructive comments. All the suggestions were considered and were included into the revised manuscript. Our manuscript has been substantially improved because of the modifications. We have made the changes and the correction suggested by you using the “Track Changes” function in the manuscript. An itemized point-by-point response to the reviewer’ comments is presented below.
Editor´s and Reviewers´ comments:
Response to Reviewer 1
RV: Reviewer
AA: Authors
|
RV: In “The usefulness of the massive open online courses postural and technological adaptations to enhance academic performance and empathy in health sciences undergraduates,“ the authors study the impact on students who use MOOC to supplement their learning versus those that do not. They found a positive impact of MOOC participation on academic performance and empathy. Those students motivated to take the MOOC were more likely to complete the MOOC option. This work contributes a valuable case to demonstrate positive outcomes to MOOC participation in a quantitative manner, which is limited in the field. There is also value in showing that empathy improves through taking a MOOC, a skill that some may be skeptical of being achieved through an online course. I do have some concerns and suggestions for revision. I am concerned about the quasi-experimental design of the study and conclusions as the authors list as a limitation of the study. If I understand it correctly, they gave students the option to participate in the MOOC to enhance their studies. So the students who did participate in the MOOC did better. The two populations (using MOOCs versus not using the MOOCs) are not random or equal presumably in terms of motivation and academic achievement before the intervention. To put it another way, if the authors just simply asked the students if they would want to participate in a MOOC with additional materials, would they see the same result for the academic performance of those who said yes versus no without the students even using the materials? They are self-selecting, which seems like a major bias to see the results as hypothesized for academic performance. Those who opt in to do more will do better.
|
|
AA: First, authors want to thank the modifications suggested by the reviewer and his/her effort to improve our manuscript. The authors are aware that a randomized clinical trial would be the best design for this research. The Quasi-experimental design was selected because the proposed MOOC could not be considered as a compulsory activity. In addition, we wanted to explore motivation and empathy without the context of compulsion. Results confirm a positive impact of MOOC participation on academic performance and empathy. The activity has been presented as optional and not compulsory. Following the reviewer´s recommendation, we have added a new sentence trying to justify our limitation. A new sentence was added in line 470 to 471 on page 12 in the revised clean version in the limitation section: “Second, there was no randomization in the assignment of participants to groups”. |
|
RV: The authors state, “Another problem that MOOCs present is that because they are very new it is not known with certainty the type of tools that are the best used to evaluate them [1].” I would not consider the xMOOCs very new. They have been around since at least 2011 and have a lot of research published about them. Beyond that, there are decades of research involving online courses or blended learning that pull in concepts similar to this paper. |
|
AA: First, authors want to thank the modifications suggested by the reviewer. Totally agree with you. Regarding recommendation the paragraph was rewritten to better understand in line 52 to 53 on page 2. “Another problem that MOOCs present is that it is not known with certainty the type of tools that are the best used to evaluate them.” |
|
RV: The introduction should be re-written to be much more sophisticated in the area of MOOC research. They must add in the papers tying together the topics of MOOCs and use with health science students, academic performance, empathy, or motivation that the authors say do not exist, but many examples are tying MOOCs to any of these topics on their own but maybe not in that complete combination. |
|
AA: First, authors want to thank the modifications suggested by the reviewer and his/her effort to improve our manuscript. Following the reviewer suggestion, we have re-written the introduction to be more sophisticated in the area of MOOCs. We have eliminated a paragraph. Several new paragraphs have been added in the introduction. Firstly, a new paragraph was added in lines 36 to 45 on page 01 in the revised clean version in the introduction section:
“Massive open online courses (MOOCs) are intended to be: (1) “massive” because thousands of students can access them, (2) “open” because participants do not pay fees, (3) “online” because they are offered through the Web, and (4) “courses” because they are shaped around specific learning objectives by offering structured contents” [1]. As McAuley put it, “a MOOC integrates the connectivity of social networking, the facilitation of an acknowledged expert in a field of study, and a collection of freely accessible online resources”[2]. MOOCs are considered very useful in increasing the educational levels of people who make use of them [3–5]. They are based on online, are free of charge and use distance training that can include videos, exercises, presentations and evaluations [3–5]”.
Secondly, a new paragraph was added in lines 55 to 64 on page 02 in the revised clean version in the introduction section:
“Numerous scientific publications based on MOOCs have been published in recent years. Many of them have focused on health science students[5–7]. A review from 2014 suggest that MOOCs can be used as a way to provide continuous medical education[8]. It also shows the potential of MOOCs as a means of increasing health literacy among the public.Recent study confirmed that students' academic performance can be influenced by MOOC which has the advantage of facilitating the learning process through offering materials and enabling the share of information [9]. Motivation is another aspect that has been briefly studied in the use of MOOCS, especially in health science students[10].Finally, the concept of empathy has been addressed in some studies with promising preliminary results[11].” |
|
RV: Overall, the writing should be thoroughly checked for grammar and clarity. I found many problems, but here are a few specific suggestions.
|
|
AA: First, authors want to thank the modifications suggested by the reviewer and his/her effort to improve our manuscript. Following the reviewer’s recommendation, we have improved our manuscript. Thank you for your comments. |
|
RV:I find the term b-MOOC odd because it is rarely used. If the authors are going to use it, they should cite an original definition and not their work using the term.
|
|
AA: Following the reviewer suggestion, we have eliminated “x-MOOCs” in the revised version. In addition to this, we have added a new paragraph in lines 36 to 45 on page 01 in the revised clean version in the introduction section:
“Massive open online courses (MOOCs) are intended to be: (1) “massive” because thousands of students can access them, (2) “open” because participants do not pay fees, (3) “online” because they are offered through the Web, and (4) “courses” because they are shaped around specific learning objectives by offering structured contents” [1]. As McAuley put it, “a MOOC integrates the connectivity of social networking, the facilitation of an acknowledged expert in a field of study, and a collection of freely accessible online resources”[2]. MOOCs are considered very useful in increasing the educational levels of people who make use of them [3–5]. They are based on online, are free of charge and use distance training that can include videos, exercises, presentations and evaluations [3–5]”.
|
|
RV: Remove calling the MOOC x-MOOCs all the time. The x versus c is rarely used.
|
|
AA: Regarding calling the x-MOOCs, they have been eliminated in the revised version. Thank you for your suggestion.
|
|
RV:The abstract and first two paragraphs of the introduction are very choppy with short sentences. The sentences don’t flow well and sometimes repeat with information that contributes little to the point of the study. The authors mostly use passive voice instead of active voice in the abstract. |
|
AA: According to the reviewer’s suggestion, the abstractand introduction has been re-written in the revised version.
|
|
RV: In the abstract, this line provides little value. “This research was based on x-MOOCs that have more classical form.” Replace this line with specifically what type of course the authors used and why. |
|
AA: Following the reviewer’s recommendation, we have improved our abstract. Thank you for your comments. |
|
RV:Missing “that” in line 44. |
|
AA: Thank you for your correction. It has been added.
|
|
RV:Line 46, should be “MOOCs” |
|
AA: We agree with you.Thank you for your correction.
|
|
RV:Line 86 does not make sense grammatically. |
|
AA: This sentencehas been corrected. Thank you for your correction.
|
|
RV: Table 3 is problematic in the formatting. |
|
AA: We believe that table 3 must have been moved when sending the document in word format. Since the format is the same for all the tables in the document. Please confirm if this is the case, so that we can modify it. Thank you |
Please, do not hesitate to contact me, if you require further corrections and information.
Thank you in advance

Reviewer 2 Report
Dear authors:
This research is properly, but conclusions must be extended and you must pay attention to formal issues: for example from line 343 to line 368 you the line spacing is different from the rest of the text. After these changes, from my point of view, this text could be published.
Author Response
ITEMIZED LIST OF THE REVIEWERS COMMENTS
Reference: [ijerph-1281527]
Title: Postural and technological adaptations of massive open online courses in pediatrics: Utility in improving academic performance and empathy
Dear Reviewer,
The authors want to thank your thoughtful and constructive comments. All the suggestions were considered and were included into the revised manuscript. Our manuscript has been substantially improved because of the modifications. We have made the changes and the correction suggested by you using the “Track Changes” function in the manuscript. An itemized point-by-point response to the reviewer’ comments is presented below.
Editor´s and Reviewers´ comments:
Response to Reviewer 2 (ijerph-1281527)
RV: Reviewer
AA: Authors
|
RV: This research is properly, but conclusions must be extended and you must pay attention to formal issues: for example from line 343 to line 368 you the line spacing is different from the rest of the text. After these changes, from my point of view, this text could be published. |
|
AA: First, authors want to thank the modifications suggested by the reviewer and his/her effort to improve our manuscript. Formatting issues have been corrected. Thank you for your suggestion.
|
Please, do not hesitate to contact me, if you require further corrections and information.
Thank you in advance

Reviewer 3 Report
General comments:
This study conducted an empirical analysis of how online education through x-MOOC affected the learning motivation, academic performance, and empathy in health science undergraduates. To that end, the authors used t-test and structural equation analysis.
This paper has implications in that it empirically analyzes the effect of education using MOOC on empathy and academic performance, which are essential for health science undergraduate students. However, there are some areas that need to be improved in the following aspects, and it is judged that the next round of review is possible only after these revisions are done.
Major comments:
1. The contents of the introduction are not systematically described and segmented according to related topics such as MOOC, EI, and technology. Before describing the research purpose, it is necessary to clearly and structurally describe the research background and gap filling. (line 35-80)
2. In 2.4 Outcome Measures, the authors did not present detailed demographic statistics about the survey respondents, the questionnaire items, and statistics on the reliability and validity of the questionnaire items. (line 147-177)
3. Both the statistics of the pre-analysis required for the mean difference test between the treatment group and the control group and the goodness-of-fit indexes with recommended criterion required for the analysis of structural equations should be presented. In addition, the authors must suggest the confirmatory factor analysis results. It seems that each empathy constructs has too many items (9, 11) and other constructs use different scales. (line 179-215)
4. It is necessary to organize the empirical analysis results more structurally.
4.1 It is recommended to present the preliminary analysis results using tables. (line 218-230)
4.2 x-MOOC and empathy: a pre-post analysis result is difficult to accept. Because, when calculating the coefficient of variation (CV), the post-group score's distribution is too wide, so the argument that x-MOOC is effective is not convincing. (line 255-259)
4.3 Are bivariate analyses necessary? Since statistical test of causality is performed through structural equation analysis, bivariate analyses are judged to be unnecessary. And check the format of Table 3. It seems that something is wrong. (line 261-285)
4.4 The structural equation results in Figure 1 need to be rewritten. Factor loadings and path coefficients should be clearly distinguished and statistical significance of path coefficients should be indicated. And Academic Performance, Cognitive Empathy, and Affective Empathy seem to use the total sum scale. What is the rationale for this? Arbitrary analysis by researchers without a rationale should be avoided. (line 287-328)
5. The Discussion part needs to be rewritten. Since the statistical analysis results are redundant, the contribution of this study should be clearly described, focusing on theoretical and practical implications. (line 330-417)
6. Because the conclusion part is too short, it needs to be rewritten considering the discussion part. (line 428-433)
Author Response
ITEMIZED LIST OF THE REVIEWERS COMMENTS
Reference: [ijerph-1281527]
Title: Postural and technological adaptations of massive open online courses in pediatrics: Utility in improving academic performance and empathy
Dear Reviewer,
The authors want to thank your thoughtful and constructive comments. All the suggestions were considered and were included into the revised manuscript. Our manuscript has been substantially improved because of the modifications. We have made the changes and the correction suggested by you using the “Track Changes” function in the manuscript. An itemized point-by-point response to the reviewer’ comments is presented below.
Editor´s and Reviewers´ comments:
Response to Reviewer 3 (ijerph-1281527)
RV: Reviewer
AA: Authors
|
RV: General comments: This study conducted an empirical analysis of how online education through x-MOOC affected the learning motivation, academic performance, and empathy in health science undergraduates. To that end, the authors used t-test and structural equation analysis. This paper has implications in that it empirically analyzes the effect of education using MOOC on empathy and academic performance, which are essential for health science undergraduate students. However, there are some areas that need to be improved in the following aspects, and it is judged that the next round of review is possible only after these revisions are done. Major comments: 1. The contents of the introduction are not systematically described and segmented according to related topics such as MOOC, EI, and technology. Before describing the research purpose, it is necessary to clearly and structurally describe the research background and gap filling. (line 35-80) AA: First, authors want to thank the modifications suggested by the reviewer and his/her effort to improve our manuscript. Following the reviewer suggestion introduction was re-writen to be more sophisticated in the area of MOOCS. We have eliminated a paragraph.A new paragraph was added in lines 59 to 68 on page 02 in the revised version in the introduction section: “Numerous scientific publications based on MOOCs have been published in recent years. Many of them have focused on health science students[5–7]. A review from 2014 suggest that MOOCs can be used as a way to provide continuous medical education[8]. It also shows the potential of MOOCs as a means of increasing health literacy among the public. Recent study confirmed that students' academic performance can be influenced by MOOC which has the advantage of facilitating the learning process through offering materials and enabling the share of information [9]. Motivation is another aspect that has been briefly studied in the use of MOOCS, especially in health science students[10].Finally, the concept of empathy has been addressed in some studies with promising preliminary results[11].” RV: Major comments: 2. In 2.4 Outcome Measures, the authors did not present detailed demographic statistics about the survey respondents, the questionnaire items, and statistics on the reliability and validity of the questionnaire items. (line 147-177) AA: Following the reviewer’s recommendation, we have added information about item sample and validity. Reliability indices for the study sample were noted on each instrument. In addition, details on the sociodemographic variables are included in section 2.1 (participants). 3. Both the statistics of the pre-analysis required for the mean difference test between the treatment group and the control group and the goodness-of-fit indexes with recommended criterion required for the analysis of structural equations should be presented. In addition, the authors must suggest the confirmatory factor analysis results. It seems that each empathy constructs has too many items (9, 11) and other constructs use different scales. (line 179-215) AA: We appreciate your suggestions. The pre-analysis required for the mean difference test between the treatment group and the control group has been incorporated into the manuscript (section 3.1.). The goodness-of-fit indexes with recommended criterion required for the analysis of structural equations are in section 3.5. Effectively each empathy construct has been supported by previous confirmatory analyzes (it has been included). 4. It is necessary to organize the empirical analysis results more structurally. AA: Without a doubt, their recommendations help us to give more clarity and structure to the manuscript. 4.1 It is recommended to present the preliminary analysis results using tables. (line 218-230) AA: Totally agree with you. Table with preliminary analysis results has been added (section 3.1.) on page 6 in the revised clean version. 4.2 x-MOOC and empathy: a pre-post analysis result is difficult to accept. Because, when calculating the coefficient of variation (CV), the post-group score's distribution is too wide, so the argument that x-MOOC is effective is not convincing. (line 255-259) AA: Effectively, you are right. Score’s distribution is too wide. However, we think that, at least in part, the MOOC is effective because the mean obtained in the post is significantly higher than the pre and the effect size is high. 4.3 Are bivariate analyses necessary? Since statistical test of causality is performed through structural equation analysis, bivariate analyses are judged to be unnecessary. And check the format of Table 3. It seems that something is wrong. (line 261-285) AA: Thank you very much for your interesting reflection. We understand your approach. The reason we include it is because a large part of the published work that analyzes structural equations does so. Although it may be a bit redundant, the truth is that it is a previous step in which it is shown if there is a relationship between the variables that will later be related in the hypothetical model. Also, Table 3 has been revised. 4.4 The structural equation results in Figure 1 need to be rewritten. Factor loadings and path coefficients should be clearly distinguished and statistical significance of path coefficients should be indicated. And Academic Performance, Cognitive Empathy, and Affective Empathy seem to use the total sum scale. What is the rationale for this? Arbitrary analysis by researchers without a rationale should be avoided. (line 287-328) AA: Totally agree with you. Effectively, factor loadings and path coefficients were not clearly distinguished and statistical significance of path coefficients should be indicated. This has been mainly due to the fact that by mistake we forgot to write the statistical significance of path coefficients. The Figure 1 has been rewritten in this direction. On the other hand, we used the total sum scale of the Academic Performance, Cognitive Empathy, and Affective Empathy in line with previous studies (i.e.: Esteve, Ramírez-Maestre & López-Martínez, 2011; Kim, Lee, Cho, & Kim, 2019; Ruiz-Párraga & López, 2015). So, all variables at the same level in the model (in this case, the result variables) have the same structure. 5. The Discussion part needs to be rewritten. Since the statistical analysis results are redundant, the contribution of this study should be clearly described, focusing on theoretical and practical implications. (line 330-417) AA: Following the reviewer’s recommendation, we have improved our discussion part. Thank you for your comments. 6. Because the conclusion part is too short, it needs to be rewritten considering the discussion part. (line 428-433) AA: Without a doubt, their recommendations help us to give more clarity and structure to the conclusion part. |
Please, do not hesitate to contact me, if you require further corrections and information.
Thank you in advance
